# Implications for Sustainability Accounting and Reporting in the Context of the Automation-Driven Evolution of ERP Systems

**Valentin Florentin Dumitru [1,\*], Bogdan-Ștefan Ionescu [1] , Sînziana-Maria Rîndașu [1] ,
Laura-Eugenia-Lavinia Barna [2] and Alexandru-Mihai Crîjman [2]**

[1] Management Information Systems Department, Bucharest University of Economic Studies,
010374 Bucharest, Romania; bogdan.ionescu@cig.ase.ro (B.-Ș.I.); sinziana.rindasu@cig.ase.ro (S.-M.R.)

[2] Doctoral School of Accounting, Bucharest University of Economic Studies, 010374 Bucharest, Romania;
barnalaura15@stud.ase.ro (L.-E.-L.B.); crijmanalexandru13@stud.ase.ro (A.-M.C.)

[\*] Correspondence: valentin.dumitru@cig.ase.ro

**Abstract:** This paper delves into the impact of the automation-driven evolution of enterprise resource planning systems (ERPSs) on sustainability accounting and reporting and the associated challenges. By employing a holistic approach focusing on the current characteristics of both ERPSs and automation platforms and an inductive approach to perform a content analysis, this study highlights how the new generation of ERPSs can support the improvement of accounting in general and sustainability accounting and reporting quality in particular. The analysis was based on information provided by the developers of leading ERPSs and automation platforms with a significant worldwide market share. In this paper, we deepen the understanding of the role that ERPSs play in improving sustainability accounting and reporting, focusing on both the benefits and challenges derived from the impact generated by increasing the integration of robotic process automation and intelligent process automation solutions. The results obtained have academic and managerial implications, addressing a research gap concerning the understanding of the consequences of ERPSs evolution from the perspective of professionals and the competitive edge companies can take advantage of to improve sustainability reporting and accounting processes.

**Keywords:** ERP systems; robotic process automation; intelligent process automation; sustainability reporting; sustainability accounting; artificial intelligence





## 1. Introduction

Today, sustainability accounting and reporting (SAR) represents an important topic for organisations, as determined by the need to comply with regulations [1–3] and increase transparency to meet stakeholders' expectations. Although a significant body of research has connected accounting with the developing idea of sustainability from the early 1990s [4], the evolution of SAR was relatively slow until the Fourth Industrial Revolution (Industry 4.0), when new technologies managed to bring more capabilities that could be successfully leveraged by organisations [5–7]. Since, in the Industry 4.0 era, companies used the information produced by the accounting information systems and financial modules available in enterprise resource planning systems (ERPSs) for SAR [8], and some companies still continue to navigate the Fourth Industrial Revolution, in the Industry 5.0 era, developers of ERPSs and automation platforms are stepping up their game to provide new solutions that will be able to respond to the needs of SAR while creating an intelligent business environment by leveraging automation solutions. Starting from this reality, the main objective of the present research is to analyse the directions of ERPSs evolution.

The development of ERPSs in past decades was driven by companies' need to overcome specific limitations. According to the definition provided in [9] (p. 1000), the role of ERP systems concerns "the integration of business processes within an organisation"

and the improvement of the way activities are performed. Therefore, in view of the constant increase in the volume of information and the obsolescence of some of the software programs used, ERPSs play an important role in the process of integrating and unifying information from multiple sources. If, in the past, the evolution of ERPSs was relatively slow, influenced by companies' need to integrate several processes within a single computer program [10], the fourth generation of ERPSs seems to be characterised by the integration of solutions intended for automation, such as robotic process automation (RPA) and automation solutions based on machine learning and artificial intelligence (AI), referred to in the specialised literature as intelligent process automation (IPA) [11,12] or intelligent robotic process automation [13,14]. Taking into account the above, analysing the evolution of ERPSs in recent years would be of interest to the readers of this journal.

The first step taken in this research to achieve the objective of the paper was to identify the main characteristics of ERPSs from the perspective of automation by analysing the functionalities and development directions of the main ERPSs that have significant market shares. This approach allowed the identification of the current state of development and an examination of how the major ERPSs' manufacturers are integrating RPA/IPA solutions that lead to improved SAR. Subsequently, the responses of the main ERP system manufacturers to the demands of the business environment were examined through a content analysis of reports and technical documentation. Finally, we examined the characteristics and functionalities of the main automation platforms through a longitudinal analysis to identify how competition between the main manufacturers causes changes in strategy and the most important functionalities introduced in terms of RPA and IPA that can facilitate SAR improvement.

Although the benefits of integrating these new emerging technologies have been confirmed both in the specialised literature and by experts in the field [15–18], recent research linking SAR with ERPSs is more focused on the impact of integration [19], which can differ from organisation to organisation. Therefore, due to the lack of engagement with a holistic approach that would take into account the range of features provided by the new generation of ERPSs, information on the impact of automation on the evolution of ERPSs, in terms of how they can improve the quality of SAR, is still vague as a result of the complexity and evolution of both ERPSs and automation-based solutions. Our study contributes with a new perspective to the scarce literature in the domain.

The present research aims to develop the level of understanding of the impact of automation solutions on the evolution of ERPSs as enablers of the improvement of SAR. To fulfil the objective, the paper is structured with a logical approach, reviewing specific aspects regarding how RPA/IPA solutions determine the evolution of ERPSs in the context of SAR, establishing and presenting the research methodology, highlighting the results obtained and the discussion, and, finally, assessing the importance of the topic and discussing the limits and conclusions.

## 2. Literature Review

SAR has become a necessary framework for "measuring techniques, and reporting the actual status of the variables in the public reports by a company" [5] due to the progressive increase in the sustainability-related requirements of different organisations, which have led to a "radical transformation of the economy and all activities" [6]. In this context, Industry 4.0 allows companies to improve their corporate environmental performance by improving SAR tools [20]. As the majority of the companies will transition to the new generation of ERPSs to meet reporting requirements and transparency expectations, it seems that early adopters can be expected to achieve superior environmental performance, such as improved material efficiency, energy savings, and a better corporate image, while late adopters may miss out on some of these benefits [21].

By using robotics, artificial intelligence, and machine learning, factories can more efficiently collect data in various areas (e.g., power, raw materials, energy constraints), thus allowing the development of strategies to mitigate constraints [22–25]. While large compa-

nies seem to benefit from strategies that can lead to substantial reductions in manufacturing costs and the development of a more competitive and stable business environment, in the case of small and medium-sized enterprises (SMEs), the challenges seem to be more frequent [26].

The relevant literature highlights that some of the main technological and human-related challenges that hamper improvements in SAR quality include the lack of adequately prepared human resources, implementation costs, the need to manipulate and analyse both unstructured and structured data, the need to have access to real-time data, the lack of employee motivation, compatibility issues, and the lack of a flexible IT infrastructure [27–31]. In this study, our aim is to present how the new generation of ERP systems can overcome these challenges with the integration of RPA/IPA solutions.

Previous research has highlighted that, from their genesis to the present, ERP systems have evolved as a result of internal and external factors, and it is now necessary for organisations to consider transitioning to systems that integrate RPA/IPA solutions and other AI-based solutions [10]. Practitioners and experts in the field, taking into account the evolution of ERP systems, have proposed the use of new terminology, the digital operations platform (DOP) [32], to reflect their capabilities more appropriately and make the distinction between traditional systems and modern solutions [33]. The rise in popularity of cloud-based solutions seems to be one of the main causes that have led to changes in the evolution of ERP systems due to their low costs, speed and accessibility, even for SMEs [34].

Although the popularity of RPA solutions is significant, the specialised literature does not provide a clear definition due to their complexity. Syed et al. [35] concluded, after a meta-analysis of the relevant specialised literature, that the purpose of RPA solutions is to provide business processes, including IT services, with two different perspectives regarding their nature: software products developed based on a set of rules, which may be, respectively, complex or advanced; and flexible software products resulting from machine learning-based training with data. The transition from relatively simple RPA solutions, which used macro commands to carry out processes that required low professional reasoning, to applications that include elements of AI (IPA), capable of undertaking more complex tasks [36], leads to significant changes within organisations, a review of internal controls being necessary to ensure increased benefits and reduced risks [37]. Asatiani et al. [38] identified three key decisions for managers regarding the adoption of RPA solutions: the type of resources that will be used (internal or external), the implementation method (local or cloud), and the type of solution (open-source or proprietary).

The success of the integration of automation solutions within accounting processes is directly influenced by the existence of a set of mechanisms that ensure compliance and risk reduction. As the existence of clear governance policies is one of the elements that contribute to ensuring the successful adoption of automation solutions, new approaches are needed to understand the resulting risks and necessary changes to internal controls [39]. Based on a systematic review of the literature and interviews with experts in the field, Plattfaut et al. [40] proposed a set of critical success factors for RPA solutions. The results obtained demonstrate that the success factors are determined by a series of components related to the organisational environment and the specifics of the project. The main categories of elements that contribute to the success of RPA solutions [40] (p. 6) are: (1) development of automation solutions in a way that allows scalability and subsequent migration to other alternatives; (2) adequate documentation by integrating knowledge from experts involved in the process and adequate knowledge management; (3) development of the necessary skills; (4) support from management; (5) the active participation of interested parties; (6) communication; (7) a strategic approach; (8) coordination in the effort optimization process; and (9) the existence of appropriate structures, including governance.

Similarly, Flechsig et al. [41] identified two categories of general barriers to RPA solutions: technical barriers, which include elements related to IT infrastructure and human resources in IT departments; and organisational barriers, such as internal communication, financial resources, management support, and organisational structure. Analysing the chal-

lenges associated with implementing an RPA solution within a multinational oil company, Fernandez and Aman [42] discovered three main categories of challenges: data security and privacy, system errors, and implementation errors.

The increase in the adoption of RPA/IPA solutions and the development of the level of understanding of the associated benefits and challenges thus lead to more realistic expectations from companies, which have begun to realise that effective use is determined not only by adoption of these solutions in such a way as to completely eliminate human resources but through collaboration between the two types of resources. Cabello Ruiz et al. [43] proposed the use of asynchronous or synchronous collaboration of resources to overcome challenges. However, in the context of SAR, Tiwari and Khan [5] emphasised the importance of a longer maturity path, as practitioners still remain cautious.

## 3. Materials and Methods

The main objective of the present research was to analyse the directions of ERPS evolution.

### 3.1. Research Design

To achieve the research objective, we conducted qualitative cross-sectional archival research by analysing the annual reports, strategies, product documentation, and review comments provided by leading ERPSs' and popular automation platforms' developers. The current study was designed within an interpretivism research paradigm, as this approach leads to the creation of "new, richer understandings and interpretations of social worlds and contexts" [44] (p. 149), which is a suitable research approach when investigating different business realities, such as those generated by the impact of automation expansion on the transformation of ERP systems. As the central objective of this study was to improve SAR in the context of ERP systems' evolution as determined by the rise of automation solutions, this qualitative method offered an efficient way to conduct an in-depth analysis. In terms of the approach deployed for theory development, the research is inductive, as the objective of this paper is to explore a phenomenon and generate new knowledge based on the data collected. Figures 1 and 2 present the study design.

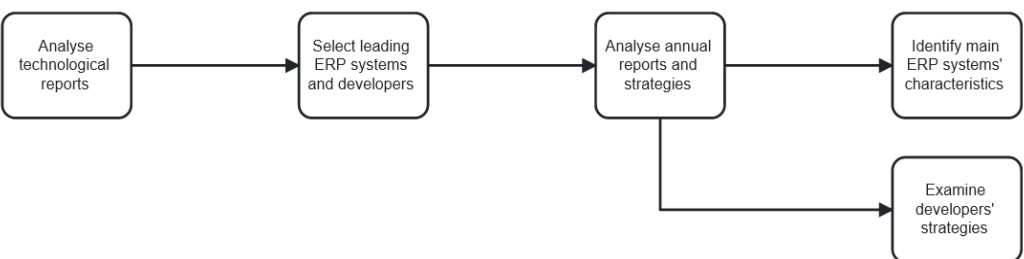

**Figure 1.** ERP systems and analysis of leading developers. Source: authors' construction.

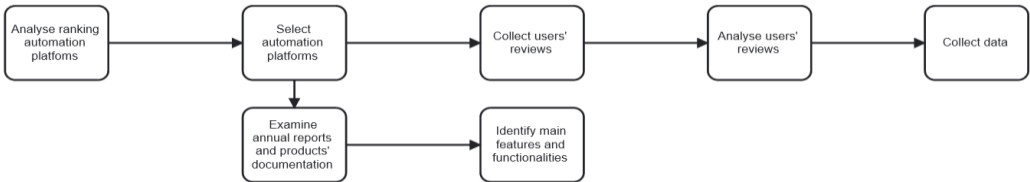

**Figure 2.** Automation platforms and analysis of leading developers. Source: authors' construction.

### 3.2. Selecting the Leading ERPSs and Their Developers

The selection of the leading ERPSs and their developers was based on the studies carried out by two independent research and consulting companies with a high level of confidence: Gartner and Forrester. These companies have examined the evolution of ERPSs in detail and promoted the progression from traditional technologies to respond to the current needs of the business environment, with an emphasis on the integration of

RPA/IPA solutions. The developer selection was based on the latest reports issued by the two companies [45–47] focused on examining Cloud ERP and DOP solutions.

Gartner uses an approach based on the completeness of vision and the ability to execute, classifying companies and ERP systems into four quadrants: leaders, visionaries, niche players, and challengers. Similarly, Forrester uses a methodology that includes 26 criteria focused on three main dimensions: market presence, strategy (performance of future products, ability to present a clear strategy, engagement, and strategic partnerships), and current offering (performance of current solutions, ease of implementation, maintenance, and implementation and training). In terms of classification, Forrester also ranks developers in four main categories: leaders, strong performers, contenders, and risky bets.

We selected ERPSs and developers categorised as leaders and visionaries/strong performers to achieve the research objective. Taking into account the methodologies used by Gartner and Forrester, leaders are represented by companies that have a strong strategy and manage to respond to market needs. Visionaries/strong performers have a complete view of the market evolution, thus demonstrating the potential to become leaders in the near future. Therefore, the developers and ERP systems analysed were:

- Oracle Corporation and Oracle Fusion Cloud ERP and Oracle NetSuite;
- SAP SE and SAP S/4HANA;
- Infor and Infor CloudSuite;
- Microsoft Corporation and Microsoft Dynamics 365.

As Forrester presents developers separately according to ERPSs performance in different domains and Workday was included in the contender category, in the report featuring DOP systems for manufacturing and distribution, we did not select this company for analysis.

### 3.3. Selecting the Leading Automation Platforms

To analyse the characteristics and main functionalities of the automation platforms, we selected the top three companies offering RPA/IPA solutions and related platforms with significant popularity according to the peer classification available on the Gartner platform [48]. As it has assisted clients to select the optimal IT solutions for more than 40 years, Gartner is regarded as a trusted advisor [49]. Furthermore, Gartner Peer Insights are considered a valuable source of users' experience suitable for content analysis [50], as they provide a complete perspective of IT solutions. In terms of the average scores for the automation platforms, the top three choices of the users, selected for analysis in the current study, were the following:

- UiPath Platform developed by UiPath Inc.;
- Automation 360 developed by Automation Anywhere Inc.;
- Blue Prism Intelligent Automation Platform developed by SS&C Blue Prism.

The selection of these companies was based on the Gartner Magic Quadrant for Robotic Process Automation [51], as all companies analysed, along with Microsoft and NICE, are considered leaders according to the methodology previously presented. We excluded Microsoft, as it has been analysed as an ERP developer and the Power Automate platform that represents the company's cloud-first automation platform is relatively recent. It received only six reviews from the finance (non-banking) industry during the period analysed. Regarding the company NICE and its RPA solution, according to the Gartner report, it is not among the top customer options and its visibility can be improved, as it has only two reviews on the Peer Insights platform. Thus, we excluded Microsoft and NICE from the analysis of the leading automation platforms.

### 3.4. Data Collection

3.4.1. ERPSs' Current Characteristics and Leading Developers' Strategies

To explore current ERPSs' characteristics and leading developers' strategies, we analysed the annual reports and strategies of the four selected companies, along with the documentation available for the products. The annual reports analysed were related to

the period 2019–2021 for three of the developers, while for Infor, only the 2019 annual report was available, as the company decided not to continue publishing the annual reports. The content analysis focused on collecting data on the products, the new features implemented, the main functionalities, the responses to changing customer needs, and the strategy fluctuations.

### 3.4.2. Automation Platforms' Current Features, Characteristics, Development Strategies, Perceived Benefits, and Challenges

To identify the main features and functionalities offered by the selected companies, we first analysed the available annual reports along with the published documentation regarding new features and product releases. Second, to examine the main perceived benefits and challenges of automation platforms from the perspective of users, as a previous study [5] stated that practitioners still demonstrate caution, especially when it comes to AI, we performed a content analysis based on reviews collected from Gartner [48]. As the purpose of the study was to examine the impact of automation on the evolution of ERP systems in the context of SAR, we exclusively selected reviews by users in the financial field between 1 January 2020 and 17 November 2022 for analysis. The COVID-19 pandemic acted as an accelerator in the digitalisation process for companies and activities [52–54]; thus, the timeline selected for analysis also captured impressions about the new features deployed. By selecting the financial (non-banking) industry, we could consider different perceptions provided by accountants and those in other positions, such as general management, business consultants, and software developers, given that the complexity of automation platforms exceeds general accountants' skills.

When providing a review, users can evaluate the following features: product capabilities, integration and development, evaluation and contracting, and service and support. Furthermore, each reviewer can present the main perceived benefits and difficulties generated by using the platform. In addition to the written review, each user can assign a score (number of stars) between 1 and 5. The average review score is presented as a weighted average of each feature. The details of the analysed platforms are presented in Table 1.

**Table 1.** Details of the analysed platforms.

| Platform | No. of Reviews * | Analysed Reviews | Average Score |
|---|---|---|---|
| UiPath Platform | 1964 | 41 | 4.5 |
| Automation 360 | 1512 | 58 | 4.5 |
| Blue Prism Intelligent Automation Platform | 691 | 33 | 4.4 |

* As of 17 November 2022. Source: authors' processing based on data collected from Gartner Peer Insights [48].

## 4. Results and Discussion

### 4.1. ERPSs Characteristics Determined by Automation Platforms

All the selected companies are supporting the evolution of ERPSs with the help of automation solutions. Over time, developers have assumed the role of pioneers in integrating new emerging technologies as a response to the shifting needs of companies determined by technological dynamics. The examination of the annual reports and strategies highlighted that ERPSs' evolution depends on internal and external factors strongly related to the field of activity. In a highly competitive environment driven by the adoption of emerging technologies, ERPSs' developers focus on migrating from on-premise solutions to cloud-based solutions. This approach allows customers to integrate solutions more efficiently and with lower costs, increasing flexibility, and provides the ability to offer customers opportunities to use innovative solutions. Moreover, this approach is also suitable for SMEs due to the reduced necessary investment.

The visions of the developers align with the expectations of the business environment, the goal being to provide suitable ERP systems for all customers. With the motto

"Innovation at the core, with no customer left behind", SAP SE [55] is currently trying to provide innovative solutions both to customers who have decided to use SAP S/4HANA and to customers who have chosen to continue using SAP ECC 6.0. Although the company had initially announced that the support period for SAP ECC would end in late 2025, the deadline was extended for another two years, most likely because the percentage of customers who migrated to SAP S/4HANA was relatively low. However, according to the 2021 annual report, SAP SE [56] achieved an annual increase in customers of 18%, reaching approximately 18,800 customers at the end of 2021. Furthermore, the report predicted continuing growth in the number of companies that use cloud platforms for the benefits provided by RPA/IPA solutions.

Oracle, although it demonstrates a vision congruent with those of the other companies analysed, prefers, for the creation of a new generation of ERP systems, to integrate automation resulting from strategic partnerships with two of the largest developers of RPA/IPA: Automation Anywhere [57] and UiPath [58]. Through this approach, Oracle can provide customers with automation solutions in a shorter time and at a lower cost, reducing any potential activity disruption and increasing productivity in a short period. Similarly, Microsoft began to release the second wave for integrating the Power Automate platform in October 2022, which includes hundreds of new functionalities within Dynamics 365 applications [59]. The leading IPA solutions included within financial applications incorporate the use of automatic character recognition, optical character recognition (OCR), the integration of financial and operational services, and the improvement of invoicing solutions.

In the vision presented by Infor [60], the future generation of ERPSs will be focused on an integrated platform based on three main benefits: (a) ensuring integration through an application programming interface on a cloud platform, (b) facilitation of user productivity through the combination of a set of capabilities of the platform, and (c) the automation provided by the use of AI solutions.

As observed in the companies' strategies, the vision is that future ERP generations will combine RPA with AI solutions. As AI-based solutions mature, companies will aim to anticipate problems and improve the speed of response to customer needs. Infor also considers that all processes within companies can benefit from using RPA/IPA solutions; however, it emphasises that most customers are still early adopters of automation solutions. The set of solutions provided by Infor Coleman includes solutions based on natural language processing techniques and machine learning, providing a "complete ensemble of tools for creating, managing, securing, and deploying machine learning models and use cases for the enterprise" [61] (p. 6).

The benefits of combining ERP systems with IPA solutions in the context of SAR are presented in Table 2.

**Table 2.** Benefits of combining ERP systems with IPA solutions in the context of SAR.

| Infor [61] | Microsoft Co. [62] | Oracle Co. [63–65] | SAP SE [55] |
|---|---|---|---|
| Process optimisation | Automation between on-premises and cloud-based applications | Modelling structured and unstructured processes | Consolidation and manipulation of data from multiple sources |
| System-level control | Automation between new and legacy applications | Decision modelling | Handling high-volume transactions |
| Increased accuracy | Improved data connectivity | Connectivity to external applications | Digitisation across multiple applications between processes |
| Increased accuracy of AI-driven results | Native ecosystem integration | Handling high-volume transactions (up to 100 steps) | Automatic approval |
| Smart infrastructure implementation | Improved scalability | Continuous activity monitoring through machine learning solutions | Simplifying new processes |

**Table 2.** *Cont.*

| Infor [61] | Microsoft Co. [62] | Oracle Co. [63–65] | SAP SE [55] |
|---|---|---|---|
| | Intelligent process automation | Intelligent automation of repetitive processes | Improved process visibility |

Source: Authors' processing based on the data collected.

As can be noticed, these benefits can be used to improve SAR by enhancing the continuous monitoring of the environmental performance of organisations through the combination of different sources of structured and unstructured data, leading to better informed decisions and better outcomes for organisations and the environment. As SAR incorporates environmental and social sustainability variables [66], these benefits can be used to improve and maintain the essential characteristics of the required data.

*4.2. Developer Responses to Market Demands*

When examining the strategies, it can be observed that developers share similar visions regarding the evolution of and responses to market demands (Table 3). As SAP SE was one of the first companies to develop an ERPS, it has again played a pioneering role in the development of efficient and sustainable organisations. Deployed in 2020, the "REINVENT" program, which aimed to remodel the way operational activities are carried out, was expanded in 2021 to help reinvent how the whole world works in the form of an intelligent network of sustainable companies [56,67,68]. It can also be observed that SAP SE's business model remained largely unchanged throughout the analysed period and was not affected by the COVID-19 pandemic; the only change was represented by the sustainability component (carbon footprint). Unlike SAP SE, which wants to guide its customers toward an intelligent and sustainable business network, Oracle's strategy is driven by the needs of the business environment. The company presents the possibility of not being able to respond promptly to the requirements determined by technological evolution as a potential risk, despite offering solutions based on emerging technologies, such as blockchain and the Internet of Things (IoT), through cloud platforms since 2019 [69]. The role of blockchain in enabling SAR results from the ability to mitigate information asymmetries [70] and improve the environmental performance of the supply chain [71]. The IoT can be used to improve SAR as it connects objects with existing networks [72], thus addressing the gap in industrial data collection and analysis [5]. Moreover, Oracle emphasises that there could be delays in customers' accepting new technologies or delays determined by the difficulties generated by the transition to the new solutions.

**Table 3.** The developer responses to market demands to improve SAR.

| Company | Responses to Market Demands |
|---|---|
| | SAP SE [56,67,73] |
| 2019 | Develops an end-to-end platform to improve customer experience |
| | Integrates advanced data analysis functionalities |
| | Develops solutions to increase the degree of engagement of the labour force |
| | Integrates automatic process scaling |
| | Acquires companies specialising in IPA and other AI-based solutions |
| | Implements the REINVENT program |
| | Provides coaching to clients to develop differentiating capabilities |
| 2020 | Supports companies in migrating to cloud ERP systems |
| | Reduces the time needed to migrate to cloud solutions and increases flexibility |
| | Introduces the SAP Business Technology Platform (SAP BTP) to provide customers with better access to IPA solutions |
| | Extends the REINVENT project to create a network of sustainable and intelligent companies |
| 2021 | Increases the degree of flexibility for users |
| | Creates a business network to react promptly to supply chain disruptions and improve environmental performance |
| | Expands the portfolio of intelligent solutions through the acquisition of Signavio Solutions LTD |

**Table 3.** *Cont.*

| Company | Responses to Market Demands |
|---|---|
| | Infor [74] |
| 2019 | Increases the speed of deployment and migration |
| | Creates a business network |
| | Introduces automatic updates within the cloud platform |
| | Introduces automatic data refinement and collection |
| | Uses Coleman's suite of AI-powered products to provide guidance to customers |
| | Microsoft [62,75,76] |
| 2019 | Increases productivity and reinvents business processes |
| | Develops an intelligent cloud platform |
| | Introduces new functionalities to increase the efficiency of interaction between users |
| | Develops cloud solutions that allow the creation of new opportunities for customers |
| | Uses AI-based solutions for data collection and interpretation |
| | Starts a strategic collaboration with Automation Anywhere to enable customers to deploy RPA solutions within the Microsoft Azure platform |
| 2020 | Develops the Power Automate platform to enable the integration of new RPA/IPA solutions |
| 2021 | Acquires companies specialising in the field of AI |
| | Continues the development of the Power Automate platform |
| | Oracle [64,65,69,77] |
| 2019 | Acquires several companies specialised in the field of cloud platforms |
| | Facilitates the migration of customers to cloud platforms |
| | Integrates emerging technologies, such as blockchain, the IoT, AI-based solutions, and chatbots |
| | Releases the Oracle Autonomous Data Warehouse to enable customers to wield IPA solutions |
| 2020 | Starts a partnership with UiPath to offer new IPA solutions to customers |
| 2021 | Deploys easily implementable functionalities that allow customers to use advanced data analysis solutions |
| | Begins partnership with Automation Anywhere to simplify the implementation of RPA/IPA solutions |

Source: Authors' processing based on the data collected.

Although Infor's financial statements could not be longitudinally analysed, from the strategy described in 2019, similarities could be observed with the other companies assessed. However, there is still a delay in adopting IPA solutions, since most clients are just starting to use these technologies [60].

Comparing the results obtained from this analysis with the most common needs of companies when selecting an ERP, such as technological factors (the cost of deployment, user friendliness and security, need fulfilment, system quality, and data accessibility) and organisational factors (financial advantages and top management support) [78,79], it can be seen that, in general, there is an alignment with market requirements. Moreover, the companies seem to exceed these requirements and provide new sustainable networks for their customers.

*4.3. Main Features and Characteristics of Automation Platforms*

By analysing the press releases regarding new functionalities and product documentation published by UiPath and Automation Anywhere, the evolution and strategies of the platforms were examined (Table 4). Similarly, to ERP system developers, companies are focusing on migrating to cloud solutions, especially software as a service (SaaS) at the expense of other types of cloud platforms. Considering the evolution and strategies of the leading ERPSs' developers, it can be noted that automation platforms will continue to evolve in the cloud environment, thus offering many more opportunities for customers to implement automation more efficiently.

**Table 4.** Main features and characteristics in the recent evolution of automation platforms suitable for optimising SAR.

| Company | Features and Characteristics |
|---|---|
| | UiPath [58,80–83] |
| 2019 | Introduces the "Long Running Workflow" concept, which allows collaboration between human resources and unattended robots<br>Includes new functionality to enable easier development of RPA/IPA solutions<br>Introduces new solutions for digitalising documents |
| 2020 | Launches Automation Cloud to enable customers to easily migrate to an SaaS model and reduce implementation costs<br>Releases a suite of products for the development of an end-to-end automation platform<br>Enters into strategic technology partnerships for direct automation of ERP systems with SAP, Salesforce, Netsuite, and Workday<br>Continues the efforts to integrate all products introduced for the development of a platform |
| 2021 | Initiates the transfer process to automation directly in the cloud<br>Simplifies the development process by decreasing the barrier represented the level of knowledge required for developing automation<br>Introduces two new types of robots: Cloud Robots and Serverless<br>Launches a new product for automation directly via the application programming interface, the Integration Service, following the acquisition of Cloud Elements, a company that offers integrations with over 150 systems<br>Acquires the Re:infer platform to automate unstructured data from digital correspondence |
| 2022 | Introduces an online application for automation development. In this sense, it opens up the opportunity for Mac and Linux users to develop automation<br>Continues the transition to the cloud and integrates products into a single platform |
| | Automation Anywhere [84–88] |
| 2019 | Introduces a set of functionalities to support the development of complex automation and good management of an extensive suite of robots at the enterprise level<br>Releases the Interactive Forms package for supervised bot development<br>Improves the efficiency of the IQ Bots to make them the best digitalisation application<br>Improves dashboards for automation monitoring<br>Integrates software bots with chatbot-based solutions for implementing conversational automation |
| 2020 | Launches the Automation 360 platform, which represents the first steps towards the cloud and offers an SaaS version of the platform<br>Improves the support function so that company representatives have more convenient and faster access to information about multiple ERP/CRM systems |
| 2021 | Introduces more products for the development of the automation spectrum: IQ Bots, a platform for developing and managing AI models, and Bot Insights<br>Develops an integration application with Bot Store, a blueprint and template platform for RPA/IPA solutions<br>Introduces a new version of the Automation Anywhere Robotic Interface for more efficient integration of RPA/IPA solutions with ERP systems<br>Acquires FortressIQ for the development of Process Discovery/Process Mining activities |
| 2022 | Introduces a new platform for digitalising documents<br>Creates new technological partnerships and develops new templates and libraries for the automation of ERP systems<br>Signs a partnership with Shibumi to integrate the CoE Manager application to collect and manage automation ideas across companies |

Source: Authors' processing based on the data collected.

Unlike UiPath and Automation Anywhere, SS&C Blue Prism is less transparent about new product releases and improvements in the solutions offered. Blue Prism, which was taken over in early 2022 by SS&C Technologies [89], aims to combine technological capabilities to offer new innovative services to clients. The main products currently offered by SS&C Blue Prism are:

- Blue Prism Cloud: an SaaS-type platform for managing RPA/IPA solutions;
- Document Automation and Decipher IDP: solutions for digitalising documents;
- Blue Prism Digital Exchange: a marketplace-type application for trading drafts and templates of automation solutions;
- UX Builder: a platform for developing simple automation for inexperienced users;
- Decision: an application for facilitating the decision-making process based on AI—currently, neither of the other two analysed companies offers such solutions.

*4.4. User Experience with Automation Platforms*

By analysing user reviews regarding the main factors representing the main reasons for selecting a particular automation platform (Table 5), it was found that the selected solutions are offered by companies with extensive experience, as it is considered that they can guarantee the functionality and performance of the product. Although most customers consider the costs relatively high, there are other factors influencing the selection of the automation platform. The key factors and reasons for purchasing items are provided along with the review comments on the Gartner Peer Insights platform, where the reviewer can select their choices from a list of predefined elements.

**Table 5.** Key factors leading to an automation platform's selection.

| Key Factor | UiPath Platform | Automation 360 | Blue Prism Intelligent Automation Platform |
|---|---|---|---|
| Product functionality and performance | 71% | 64% | 67% |
| Strong service expertise | 59% | 53% | 39% |
| Product roadmap and future vision | 44% | 34% | 27% |
| Strong user community | 44% | 26% | 12% |
| Breadth of services | 44% | 21% | 27% |
| Strong customer focus | 34% | 45% | 21% |
| Overall costs | 27% | 24% | 24% |
| Financial/organisational viability | 24% | 24% | 30% |
| Strong consulting partnership | 20% | 19% | 18% |
| Pre-existing relationships | 12% | 19% | 9% |

Source: Authors' processing based on the reviews analysed.

RPA/IPA developers must constantly improve their solutions to respond effectively to the needs of the business environment. By establishing strategic partnerships with the leading ERPSs' developers and vendors, customers can access platforms that successfully integrate various components, resulting in increased efficiency and creating new strategic advantages. However, strategic technological partnerships do not represent a primary driver for users, although their existence improves compatibility between the solutions used, thus contributing to an increase in the quality of SAR.

As in the case of the key factors that determine the selection of automation platforms, the reasons for using a particular automation product were diverse (Table 6), and most customers appreciated the ability to stimulate innovation, improvement, and efficiency in relation to processes. Comparing the reasons with the identified challenges (Table 7), it can be observed that, although the ability to increase compliance and effective risk management was not an initial critical characteristic, users evaluated security compliance as a significant feature; in one of the review comments, a user pointed out the lack of data security compliance ("Not compliant with large scale security and risk compliance hence a major flag for continuity"). Data security and privacy represent core characteristics of both financial and nonfinancial information. Despite the fact that some financial analysts do not consider cybersecurity risks in investment analyses due to the perceived lack of usefulness [90], they represent a main pillar of ensuring sustainability [91].

Although the developers of automation platforms, similar to the ERPSs' developers, aim to create intelligent business networks to improve collaboration and address disruptions more efficiently, users did not consider this aspect a key reason for selecting a particular platform. This result may indicate that, in addition to responding to customer needs, automation platforms developers have to transform the environment as part of their strategy by pioneering the development of new business models based on intelligent collaboration between companies, thereby enhancing sustainable processes.

Users' reviews of automation platforms highlighted several difficulties (Table 7), mainly focused on the limited abilities of solutions integrated within AI-based platforms. In the context of the increased adoption of IPA solutions, developers are considering migrating from RPA solutions to IPA to respond to the needs of the business environment

and develop competitive advantages. However, users seem to have encountered challenges regarding the three platforms analysed ("We had trouble with OCR detection", "We would like to see more reliable OCR capability to have a one product solution with accuracy in all applications of this RPA solution").

**Table 6.** Reasons for purchasing a product/service.

| Reasons | UiPath Platform | Automation 360 | Blue Prism Intelligent Automation Platform |
|---|---|---|---|
| Drive innovation | 71% | 52% | 42% |
| Improve business process outcomes | 59% | 48% | 61% |
| Create internal/operational efficiencies | 59% | 43% | 64% |
| Improve business process agility | 49% | 59% | 58% |
| Drive revenue growth | 46% | 28% | 36% |
| Cost management | 41% | 31% | 36% |
| Improve customer relations/service | 37% | 26% | 33% |
| Reduce time to market | 37% | 24% | 36% |
| Enhance decision making | 29% | 17% | 12% |
| Improve compliance and risk management | 22% | 29% | 33% |
| Improve supplier or partner relationships | 17% | 16% | 6% |

Source: Authors' processing based on the reviews analysed.

**Table 7.** Main challenges encountered by users.

| Difficulties | UiPath Platform | Automation 360 | Blue Prism Intelligent Automation Platform |
|---|---|---|---|
| Algorithms not updating automatically | | ✔ | |
| Compatibility issues | ✔ | ✔ | ✔ |
| Complex initial implementation | | | ✔ |
| Connectivity issues | | ✔ | |
| Data security issues | ✔ | | ✔ |
| Difficult troubleshooting | ✔ | ✔ | ✔ |
| Frequent updates | ✔ | ✔ | |
| Functional errors | ✔ | ✔ | |
| Hidden algorithmic logic | ✔ | ✔ | |
| High costs | ✔ | ✔ | ✔ |
| Lagging processes | | | ✔ |
| Limited compatibility | ✔ | ✔ | |
| Limited OCR capabilities | ✔ | ✔ | ✔ |
| Limited or insufficient support | ✔ | ✔ | ✔ |
| Limited number of programming languages accepted | ✔ | ✔ | |
| Low customisation of interface | ✔ | | |
| Manual adjustments required | | ✔ | |
| Outdated documentation | ✔ | | |
| Prior skills required | ✔ | ✔ | ✔ |
| Reduced level of automation | | ✔ | ✔ |
| Significant differences between versions | | ✔ | |
| Updating issues | | ✔ | ✔ |

Source: Authors' processing based on the reviews analysed.

The lack of similarities between versions appears to be another critical aspect identified in the case of the Automation 360 platform, with users feeling that the developers should have considered keeping the same design ("Completely redesign from v11, it is like learning a new tool when shifting to A360"). With the significant modification of the platform, users must allocate significant resources to perform activities correctly. Therefore, these changes can lead to significant disruptions to activity in the short term.

Compatibility issues with different solutions and applications represent a challenge for all three analysed platforms, although the developers' strategy is to eliminate such situations. Even though some of the customers use less popular software, it seems that compatibility issues occur even with widely used applications ("There are few commands that are very useful but not compatible with Google Chrome"; "[ … ] poor integration with some software such as web applications"; "There are no serious hiccups that I have faced this product, but one improvement area can be integration capability with the newer tools in the market").

## 5. Conclusions

This research aimed to examine the impact of automation on the evolution of ERPSs and how this evolution can improve SAR. Using a holistic approach, we explored the main characteristics of ERPSs through the lens of RPA/IPA, showing how ERP systems' developers respond to market demands and examining the main characteristics and functionalities of automation platforms that can be leveraged to improve SAR. Although ERP systems' and automation platforms' developers are improving the solutions needed to address some of the challenges and contribute to the improvement of SAR, there are still some limitations.

The results obtained highlight that the evolution of ERP systems is determined by internal and external factors. When analysing the strategies of the ERP systems' developers, two different perspectives were observed: the need to respond to the current customers' requirements, regardless of the degree of adoption of technologies; and the assumption of a pioneering role in the development of intelligent and sustainable business networks, thus making it possible to successfully overcome the disruptions that may occur within business processes. Currently, ERP systems are turning into DOP platforms that integrate traditional ERP systems with IPA, thus facilitating and improving SAR. By migrating to DOP cloud platforms, developers offer companies the opportunity to use innovative solutions in a timely manner with reduced implementation costs, providing a more viable solution for SMEs. By creating strategic technology partnerships with the developers of automation solutions or creating RPA/IPA solutions, companies can build intelligent platforms that can be successfully used in any industry and improve the accuracy and transparency regarding the environmental performance.

Automation platforms' developers are taking a similar approach, investing significantly in cloud platforms and creating strategic partnerships with ERPSs' vendors. Customers' perceptions of the main benefits and associated challenges of using the leading automation platforms highlight that the benefits are significant. It seems that only by leveraging the synergies between ERPSs' and automation platforms' developers can companies improve the quality of SAR. However, there are also several challenges, as RPA/IPA solution developers cannot easily integrate different IT solutions.

The results of this study indicate that we cannot currently discuss about a new, well-defined generation of ERPSs completely focused on SAR due to the increased complexity of current emerging technologies. Thus, in the current context, although similarities can be observed, especially regarding the future use of DOP platforms, which can act as great enablers for SAR, not all companies are easily joining in this transition, as can be seen in the case of SAP SE, which has continued to extend the deadline for the discontinuation of support for SAP ECC 6.0. Most companies already use several automation solutions, but, as users' perceptions demonstrate, extensive use of RPA/IPA requires a solid knowledge set to employ these solutions effectively. Therefore, this study contributes at improving the understanding regarding the automation-driven evolution of ERP systems through RPA/IPA and its impact on improving SAR, taking into account the visions of developers, the perceptions of customers, and their degree of acceptance of the transformation of processes.

As we only analysed companies that are leaders or have the potential to become leaders, we did not examine how other ERP systems' developers respond to the changes determined by the evolution of automation. However, the selected companies have shown

that capitalising on technological progress and becoming pioneers can contribute significantly to developing competitive advantages. Another limitation of this study is that it focused primarily on factors that can improve SAR. However, in response to the general complexity of ERPSs and automation platforms, this study highlights key factors related to the automation-driven evolution of ERPSs. Future research could continue to longitudinally analyse changes in strategy and the main key factors driving the developers of ERP systems and RPA/IPA platforms to examine whether company objectives prevail over the demands of the business environment.

This research has several important implications for both academia and business. For academia, it provides a glimpse of how universities can respond to technological changes within financial accounting processes, preparing future practitioners to successfully use DOP for SAR and providing explicit knowledge about the impact generated by the integration of RPA/IPA solutions. From a managerial perspective, companies should evaluate the decisions to integrate new technologies and migrate to other platforms only after careful analysis, ensuring that they have all the necessary resources and that the impact of possible disruptions can be appropriately addressed, especially in the case of SMEs. Furthermore, given the findings of this study, companies should also consider, when selecting an ERP solution, the degree to which that product can help them optimise their level of performing accounting and reporting activities more sustainably and its overall contribution to the general development of the company. As emerged from the user perceptions investigation, the automation platforms are complex and, in addition to the obvious benefits, additional resources are required to overcome several challenges and thus continue to improve SAR.

**Author Contributions:** Conceptualization, V.F.D., B.-Ș.I., S.-M.R., L.-E.-L.B. and A.-M.C.; methodology, V.F.D., B.-Ș.I., S.-M.R., L.-E.-L.B. and A.-M.C.; software, V.F.D., B.-Ș.I., S.-M.R., L.-E.-L.B. and A.-M.C.; validation, V.F.D., B.-Ș.I., S.-M.R., L.-E.-L.B. and A.-M.C.; formal analysis, V.F.D., B.-Ș.I., S.-M.R., L.-E.-L.B. and A.-M.C.; investigation, V.F.D., B.-Ș.I., S.-M.R., L.-E.-L.B. and A.-M.C.; resources, V.F.D., B.-Ș.I., S.-M.R., L.-E.-L.B. and A.-M.C.; data curation, V.F.D., B.-Ș.I., S.-M.R., L.-E.-L.B. and A.-M.C.; writing—original draft preparation, V.F.D., B.-Ș.I., S.-M.R., L.-E.-L.B. and A.-M.C.; writing—review and editing, V.F.D., B.-Ș.I., S.-M.R., L.-E.-L.B. and A.-M.C.; visualization, V.F.D., B.-Ș.I., S.-M.R., L.-E.-L.B. and A.-M.C.; supervision, V.F.D.; project administration, V.F.D.; funding acquisition, B.-Ș.I. All authors have read and agreed to the published version of the manuscript.

**Funding:** The APC was funded by the Bucharest University of Economic Studies.

**Data Availability Statement:** Not applicable.

**Acknowledgments:** This research was supported by the Bucharest University of Economic Studies through the research grant "The impact of automation on the evolution of ERP systems in the context of the accounting profession".

**Conflicts of Interest:** The authors declare no conflict of interest.

## Abbreviations

| | |
|---|---|
| AI | Artificial intelligence |
| DOP | Digital operations platform |
| ERP | Enterprise Resource Planning |
| ERPS | Enterprise Resource Planning System |
| IoT | Internet of Things |
| IPA | Intelligent process automation |
| IT | Information technologies |
| RPA | Robotic process automation |
| SaaS | Software as a service |
| SAR | Sustainability accounting and reporting |
| SME | Small and medium-sized enterprises |

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
