# Peer review of "Implications for Sustainability Accounting and Reporting in the Context of the Automation-Driven Evolution of ERP Systems"

_electronics, doi:10.3390/electronics12081819_

Round 1
Reviewer 1 Report
The researchers are studying the role of Enterprise Resource Planning Systems (ERPS) in improving sustainability accounting and reporting by focusing on the benefits and challenges arising from the integration of robotic process automation and intelligent process automation solutions. The topic is relevant and worth being discussed; moreover, the authors have performed sufficient literature analyses to prove the topicality of the issue.
The methodology chosen for the research - qualitative cross-sectional archival research - should be considered appropriate; nevertheless, I would suggest the following improvements:
- To explain the algorithm of the selection of platforms (3 out of considerably greater number worldwide); the reason for selecting different numbers of reviews for analysis (2%, 4%, and 5%), as well as the criteria applied for selection and evaluation (average score – could it be that the number is biased?);
- By analyzing the developers’ responses to the market demands (Table 3), it is less clear what was the particular market demand (how it is determined and evaluated) and what functionalities were not implemented even if demanded by the users;
- Table 5 and Table 6 provide numerical information regarding the factors leading to the automation platform’s selection and reasons for purchasing the product or service; please explain the selection of the criteria, intercorrelation issue, and statistical significance and validity of the results.
The conclusions are consistent with the evidence presented in the article; nevertheless, I advise the authors to include a scientific discussion of the results gained and demonstrate the potential implications of the achievements in practice and future research.
Having said that, the article is worth being published as soon as the authors have revised it and implemented the comments mentioned above.
I invite you to clarify the title so that it is not in two sentences.
Author Response
Dear Reviewer,
We appreciate your precious time in reviewing our paper and providing valuable comments. It were your insightful suggestions and comments that led to improvements in our manuscript. We carefully considered the comments and tried our best to address each of them. Below we provide the point-by-point responses. All modifications in the manuscript are highlighted.
Regarding the algorithm of the selection of platforms we have presented the methodology in a more extensively manner. The reason for selecting a different number of reviews for each application is given by the fact that each of the analysed solutions has a specific number of reviews received from professionals in the financial industry during the period examined. Moreover, to address the issue regarding the lack of transparency in terms of the market demands, we presented the results in comparison with the key factors taken into account by companies when selecting an ERP system.
As per your recommendation, we have extended the information regarding the key factors. However, given the fact that a reviewer can select the list of factors considered appropriate in his/her case, no other statistical analysis could have been carried out with the exception of the frequency analysis presented in the study.
To address the final comment, we changed the title of the paper by having only one sentence and improving its clarity.
We hope that the new version of the manuscript will be closer to reaching a strong contribution to the literature!
Thank you again for your kind support!
The Authors

Reviewer 2 Report
Dear Sir,
After review of the paper titled "Enhancing ERP systems with automation. Implications for sustainability accounting and reporting" which submitted to Electronics Journal, I think that this paper is suitable for publication in the current form.
First, this paper delves into the impact of the automation-driven evolution of Enterprise Resource Planning Systems (ERPS) on sustainability accounting and reporting and the associated challenges.
Second, this paper presents many empirical contributions which can be used by researchers, academics and market managers.
Third, this paper is very good organized.
Finally, I think that the editor considers it for publication in the current form.
All the best
Author Response
We are very grateful for the comments, suggestions, and reflections on our paper!
Reviewer 3 Report
Dear Authors,
the paper is very well written and easy to read. The introduction explains the background and highlights the gap that the research aims to fill. The literature review is extensive and well developed, with up-to-date references. The section on materials and methods describes the approach very well. The results are clearly described and the discussion is relevant. The conclusions are appropriate and they highlight the theoretical and managerial implications of the research.
Good luck!
Author Response

(The authors gave the same response as above.)

Reviewer 4 Report
This article evaluates the impact of the automation-driven evolution of Enterprise Resource Planning Systems (ERPS) on sustainability accounting and reporting. It highlights the potential benefits and drawbacks of incorporating robotic process automation and intelligent process automation solutions into ERPS to improve the quality of sustainability accounting and reporting. The literature review provides a comprehensive overview of sustainability accounting and reporting (SAR) and Industry 4.0's impact on enhancing corporate environmental performance.
This study focuses solely on SAR, which may pose a limitation since it may not encompass all the complexities of ERPS and automation platforms' functionalities and features. The research questions are answered using a content analysis methodology that scrutinizes data from leading ERPS and automation platforms developers with a significant global market share to identify ERPS's primary characteristics in terms of automation that can enhance sustainability accounting and reporting. The article also investigates how ERP manufacturers respond to sustainability accounting and reporting market demands while examining the features and functions of automation platforms. Moreover, the manuscript solely concentrates on financial user reviews, which may not represent users in other fields. Lastly, the paper lacks empirical evidence to substantiate the claims made, and further research is necessary to confirm the findings. However, these limitations do not detract from the research's value and contribution.
The practical implications of this research are that companies can leverage the latest generation of ERPS to enhance their sustainability accounting and reporting processes. The study advises companies to carefully evaluate the decision to integrate new technologies and migrate to other platforms. Additionally, the paper proposes future research to longitudinally analyze changes in strategy and key factors driving the development of ERP systems and RPA/IPA platforms.
Overall, this research provides valuable insights into the impact of automation on sustainability accounting and reporting and the potential benefits and limitations of integrating new technologies into ERPS.
Congratulations to the authors for the research work carried out.
Author Response
Dear Reviewer,
We appreciate your precious time in reviewing our paper and providing valuable comments. It was your insightful suggestions and comments that led to possible improvements in our manuscript. We carefully considered the comments and tried our best to address each of them. Below we provide the point-by-point responses. All modifications in the manuscript are highlighted.
As per your review, we extended the limitations of the paper regarding the complexities of ERPS and automation platforms’ functionalities and features analysed solely from the perspective of SAR. By selecting only the financial (non-banking) industry when conducting the user-reviews analysis, we were able to capture more perspectives, not only those of accountants, but also managers, consultants, and software developers. As the financial industry has in general a higher understanding of the needs for sustainable accounting and reporting, we considered appropriate to select only this industry. Regarding the comment that the paper lacks empirical evidence to substantiate the claims made, and further research is necessary to confirm the findings, we would like to clarify that we have selected for analysis only the companies that are considered leaders in the ERP and automation platforms fields, taking into consideration the majority of relevant data as per the objective of this study.
We hope that, after careful revisions, the manuscript will be aligned with the standards of this respected journal and welcome further constructive comments if any.
Thank you for your valuable comments!
